# High-Density Genetic Linkage Map Construction and QTLs Identification Associated with Four Leaf-Related Traits in Lady's Slipper Orchids (*Paphiopedilum concolor* × *Paphiopedilum hirsutissimum*)

**Dong-Mei Li and Gen-Fa Zhu ***

Guangdong Key Lab of Ornamental Plant Germplasm Innovation and Utilization, Environmental Horticulture Research Institute, Guangdong Academy of Agricultural Sciences, Guangzhou 510640, China
* Correspondence: genfazhu@163.com

**Abstract:** Lady's slipper orchids (*Paphiopedilum* spp.) are highly valuable within the flower industry. Recently, both *Paphiopedilum concolor* and *Paphiopedilum hirsutissimum* (2n = 2x = 26) have been widely used for hybrid parents, ornamental, and economic purposes. However, high-density genetic maps and leaf traits related to quantitative trait loci (QTLs) in these two *Paphiopedilum* species have been poorly studied. Herein, an interspecific F1 population of 95 individuals was developed from the cross between *P. concolor* and *P. hirsutissimum* with contrasting leaf length (LL), leaf width (LW), leaf thickness (LT), and leaf number (LN). RNA extracted from the F1 population and their parents was subjected to high-throughput RNA sequencing. Approximately 745.59 Gb of clean data were generated, and were assembled into 349,730 transcripts and 185,089 unigenes. In total, 165,196 high-resolution polymorphic single nucleotide polymorphism (SNP) markers were initially identified. Finally, 8410 SNP markers satisfied the requirements and were used to construct a genetic map. The integrated map contained 13 linkage groups (LGs) and spanned 1616.18 cM, with an average distance of 0.19 cM between adjacent markers. QTL analysis in the F1 population identified 12 QTLs linked to four leaf-related traits, including LL, LW, LT, and LN. These QTLs by composition interval mapping, explained 11.86% to 21.58% of the phenotypic variance, and were distributed on eight LGs, but not on LGs 4, 6, 8, 12, and 13. Furthermore, 25 unigenes were identified via BLAST searches between the SNP markers in the QTL regions and our assembled transcriptome, of which 11 unigenes were enriched with 59 gene ontology (GO) terms. The information generated in this study will be useful for candidate genes for further molecular regulation studies on leaf traits, future marker-assisted selection of leaf ornamental improvement breeding, genome assembly, and comparative genome analyses.

**Keywords:** *Paphiopedilum*; RNA sequencing; single nucleotide polymorphism; genetic map; quantitative trait loci; leaf traits



## 1. Introduction

*Paphiopedilum* spp. are valuable ornamental plants, commonly known as lady's slipper orchids in horticulture, belonging to the genus *Paphiopedilum* family Orchidaceae [1,2]. There are about 79 species distributed mainly from southwestern China to Southeast Asia, with some species extending to Nepal, southern India, New Guinea, and Solomon Islands [3,4]. Since the 1980s, lady's slipper orchids have become a pillar of the flower industry, especially within the orchid industry [3]. Moreover, the whole plants of *P. micranthum*, *P. concolor*, *P. parishii*, *P. dianthum*, *P. insigne*, and *P. appletonianum* in China are used for the treatment of tuberculosis, snakebite, cough asthma, splenomegaly, injuries from falls, and other diseases [5].

The artificial hybridization breeding of *Paphiopedilum* has a long history of more than 150 years. As of July 2022, a total of 28,748 hybrids have been registered in the Royal Horticultural Society (http://apps.rhs.org.uk/horticulturaldatabase/orchidregister/

orchidresults.asp (accessed on 5 September 2022)). Among these selected *Paphiopedilum* hybrids, the breeders have mainly focused on the colorful and distinctive flowers, but have paid little attention to the interesting and distinctive leaves, for example, *P. concolor* has oblong to elliptic, adaxially tessellated leaves with dark green and whitish or a light green color, and *P. hirsutissimum* has lorate or linear single green leaves [3,6]. Both species have 2n = 2x = 26 [7,8]. Furthermore, the two species have different flower colors, sizes, and shapes. *P. concolor* usually has 1–3 flowers per scape, 5–7 cm across, pale yellow or yellowish, finely spotted with purple or brown-purple throughout, dorsal sepal broadly ovate, obtuse to retuse at apex, synsepal similar to dorsal sepal, slightly smaller, and petals obliquely elliptic, rounded or subtruncate at apex; while *P. hirsutissimum* has a single flower per scape, 9–13 cm across in full bloom, dorsal sepal and synsepal dark brown, with pale yellow-green margins, petals pale yellow and densely and finely spotted with purple-brown in basal half, purplish-rose in apical half, and petals strongly undulate along basal margins [3,8]. So far, 178 hybrids derived from *P. concolor* and 170 hybrids derived from *P. hirsutissimum* have been registered in the Royal Horticultural Society, respectively. Therefore, these two *Paphiopedilum* species have been intensively used for female parent or/and male parent. In *Paphiopedilum* breeding practice, the popular cultivars in the market usually have oblong and adaxially tessellated leaves, such as *Paphiopedilum* Maudiae 'Red Swan' and *Paphiopedilum* Maudiae 'Magnificum' [3]. Therefore, it is important to analyze the genetic mechanisms of leaf morphology to breed new *Paphiopedilum* cultivars with higher ornamental and economic values.

Leaf morphology is a complex trait involving leaf length, leaf width, leaf number, and so on. Most of these leaf traits are quantitatively inherited and controlled by multiple quantitative trait loci (QTLs). The construction of genetic maps and QTL analyses have been proved to be powerful tools for the identification of candidate genes associated with leaf traits in model plant *Arabidopsis thaliana* [9], many important foods, oil and vegetable crops [10–16], and woody trees [17]. However, in orchids, only *Dendrobium* species' genetic maps and QTL analyses have been intensively reported to date. The first *Dendrobium* map included 209 random amplified polymorphic DNA (RAPD) and 98 sequence-related amplified polymorphism (SRAP) markers [18]. Subsequently, with more types of molecular markers, such as expressed sequence tag-simple sequence repeats (EST-SSR) and inter-simple sequence repeat (ISSR), a number of *Dendrobium* genetic maps were constructed [19–21]. In *D. officinale* and *D. aduncum*, a genetic map was constructed based on 20 EST-SSR and 160 SRAP loci with a total distance of 1580.4 cM and a mean of 11.89 cM between adjacent markers [19]. Then, according to a total of 422 markers, including 66 EST-SSR, 126 SRAP, 74 ISSR, and 156 RAPD markers, the genetic map of *D. moniliforme* was constructed with 1127.9 cM in total length with 165 marker loci distributed in 17 linkage groups (LGs), and the *D. officinale* genetic map consisted of 19 LGs with a total length of 1210.9 cM positioned by 169 marker loci [20]. Immediately, using 286 RAPD loci and 68 ISSR loci, the genetic map of *D. nobile* was constructed with 1474 cM in total length with 116 loci distributed in 15 LGs, and the *D. moniliforme* genetic map had 117 loci placed in 16 LGs spanning 1326.5 cM [21].

In recent years, the next-generation sequencing technique can be used to detect large quantities of single nucleotide polymorphism (SNP) markers in the whole genome. Several methods have been used for the identification of SNPs, such as specific-locus amplified fragment (SLAF) sequencing [22] and RNA sequencing [23]. By using SLAF sequencing, high-density genetic maps in ornamental plants have been constructed for *Dendrobium* [24] and *Chrysanthemum* [25]. In *Dendrobium*, the genetic map for two parents (*D. moniliforme* × *D. officinale*) consisted of 8573 SLAF markers, covering 19 LGs and spanning a length of 2737.49 cM with an average distance of 0.32 cM between adjacent markers [24]. In *Chrysanthemum*, the genetic map comprised 6452 SLAF markers with an average map distance of 0.76 cM, covering 27 LGs [25]. By RNA sequencing, 9564 SNP markers were used to construct the first high-density genetic map for female *D. nobile* with 3608 cM in total length and an average marker interval of 0.41 cM, and three expression QTLs (eQTLs) related to stem length and diameter were identified [26]. Although many molecular

markers, genetic maps, and QTL analyses have been reported in *Dendrobium* species, the high-density map and QTL mapping of leaf traits in *Paphiopedilum* species are lacking.

In this study, we aimed to construct the high-density genetic linkage map and identify QTLs associated with leaf length-, leaf width-, leaf thickness-, and leaf number-related traits in *Paphiopedilum* species. The map was constructed using SNP markers developed by RNA sequencing. Additionally, 25 candidate genes in the four leaf traits-related QTL regions were uncovered. These findings will promote research on the regulation of leaf length, leaf width, leaf thickness, and leaf number in the *Paphiopedilum* species, which would be useful for breeders to increase the ornamental values in *Paphiopedilum*, and for map-based cloning and comparative genomes research.

## 2. Materials and Methods

### 2.1. Plant Materials

In the autumn of 2014, the ripe seed capsules were collected from the F1 population of an interspecific cross between species *P. concolor* (female) and *P. hirsutissimum* (male). Between September 2014 and August 2015, the F1 seed capsules were sterilized for 20 min in 0.1% (*w/v*) mercuric chloride ($HgCl_2$) and subsequently rinsed 5 to 6 times with sterilized distilled water to remove traces of $HgCl_2$. The seeds were then removed from the capsules and cultured onto one-sixth of MS medium, which was used as the basic medium, supplementing with 2.0 mg/L 6-BA in combination with 0.5 mg/L NAA, 15% coconut milk, 10% banana juice, 30 g/L sucrose, 1 g/L activated charcoal, and 0.8% (*w/v*) agar. On September 2015, the young plants were transferred to a greenhouse and grown in plastic pots (diameter, 6 cm) in Environmental Horticulture Research Institute, Guangdong Academy of Agricultural Sciences (23°23′ N, 113°26′ E), under natural light at 15 °C to 32 °C. All plant materials were watered and fertilized as needed. Finally, a total of 95 three-year-old F1 hybrid individuals with at least three leaves were randomly selected for phenotypic determination and RNA extraction.

### 2.2. Leaf-Related Traits Determination

For four leaf-related phenotypic traits, 95 three-year-old F1 individuals were arbitrarily selected for determination of leaf number of the plant, leaf length, leaf width, and leaf thickness of the second leaf from the top before RNA sample leaf collection. Leaf thickness was measured in the middle section of the leaf on the side as near to the main midrib as possible. Leaf length and leaf width were measured with a standard soft ruler, leaf thickness was measured with a vernier caliper, and leaf number was observed by counting with the naked eye. Leaf traits data were recorded using Microsoft Office Excel 2010. The descriptive statistics, frequency distributions, means, standard deviations, variances, F values, and Spearman correlation coefficients were performed using IBM SPSS statistics 16 (SPSS Inc., Chicago, IL, USA). The coefficient of variance was calculated by the following formula:

$$CV = SD/M \times 100\% \tag{1}$$

where, CV = coefficient of variance, M = mean of each trait in the F1 population, and SD = standard deviation of each trait in the F1 population.

As the F1 population was full-sibs, broad-sense heritability ($H^2$) was estimated as follows:

$$H^2 = (1 - 1/F) \times 100\% \tag{2}$$

where, F value was calculated by an analysis of variance for each trait in the F1 population in a univariate module of a general linear model in SPSS statistics.

### 2.3. RNA Extraction, cDNA Library Construction, and RNA Sequencing

Fresh second leaves from the top were collected from 95 F1 progenies, along with the female parent and male parent, respectively. Collected leaves were immediately frozen in liquid nitrogen and stored at −80 °C. Total RNA for each sample was extracted from

collected leaves using RNeasy Plant mini kit (Qiagen, Tokyo, Japan), and then treated with RNase-free DNaseI (TaKaRa, Dalian, China) to avoid genomic DNA contamination. The RNA was quantified using NanoDrop$^{TM}$ 1000 spectrophotometer (NanoDrop Technologies, Wilmington, DE, USA), whereas the integrity and quality of RNA was checked by Agilent 2100 Bioanalyzer (Agilent Technologies, Santa Clara, CA, USA). The RNA sample with OD260/OD280 > 1.8, total amount $\geq$ 0.5 µg, RNA integrity number (RIN) $\geq$ 6.9, and 28S/18S > 1 was used for follow-up cDNA library construction. cDNA library was constructed for each sample independently using the previously reported method [8], and then sequenced independently for each sample using Illumina HiSeq X Ten instrument to obtain paired-end reads of 150 bp from both ends at Biomarker Technology Corporation in Beijing, China. All sequencing data were uploaded to NCBI Bioproject with the accession number PRJNA760286.

### 2.4. RNA Sequencing Data Assembly and Annotation

All the raw paired-end reads were filtered with the Illumina read trimming tool Trimmomatic v0.39 [27] by the removal of sequencing adapters, low quality sequences, and cDNA synthesis primers. The two RNA sequencing libraries data from female (*P. concolor*) and male (*P. hirsutissimum*) leaves were united prior to assembly to generate a reference assembly. Then all high-quality clean reads were used for de novo assembly with the software Trinity v1.0 [28]. In order to evaluate the quality of the transcriptome, three different methods were used: (1) to evaluate the randomness of mRNA fragmentation and the degradation of mRNA by examining the distribution of the inserted fragments on unigenes; (2) to evaluate the dispersion of insert length by mapping the length distribution of the inserted fragments; and (3) to assess library capacity and adequacy of clean reads against the unigenes by mapping saturation. The clean data of each sample were mapped back to the assembled transcripts or unigenes by software STAR v2.4.0 (http://code.google.com/p/rna-star/ (accessed on 5 September 2022)) [29], and those mapped clean reads were used for subsequent SNP calling. The resulting transcript or unigene sequences were annotated via BLAST with E-values less than $10^{-5}$ by following publicly available databases: National Center for Biotechnology Information (NCBI) non-redundant (Nr) protein database (http://www.ncbi.nlm.nih.gov (accessed on 5 September 2022)), Swiss-Prot protein database (http://www.expasy.ch/sprot (accessed on 5 September 2022)), Cluster of Orthologous Groups (COG) database (http://www.ncbi.nlm.nih.gov/COG (accessed on 5 September 2022)), Gene ontology (GO) [30], Pfam (Protein family), and Kyoto Encyclopedia of Genes and Genomes (KEGG) Automatic Annotation Server (KAAS) (http://www.genome.jp/tools/kaas/ (accessed on 5 September 2022)) [31].

### 2.5. SNP Marker Calling and Genotyping

Mapped clean reads were used to make SNP calling between two parents and the F1 progenies using the GATK software kit v4.1.1.0 [32]. Then, a perl script was used to filter the SNPs that were missed in parents, the sequencing depth was less than 4-fold in parents, and there were no polymorphic markers between filter markers. Filtered SNPs needed to be genotyped, and the genotypic coding rule used was the universal biallelic coding rule in genetics. Polymorphic SNPs were classified into eight segregation patterns (aa × bb, ab × cc, ab × cd, cc × ab, ef × eg, hk × hk, lm × ll, and nn × np). Due to the F1 population of slipper orchids being a cross-pollinator population, all eight segregation patterns except for aa × bb were suitable for map construction.

### 2.6. Genetic Linkage Map Construction and Evaluation

To construct a high-quality genetic map, polymorphic SNPs with average sequence depths of more than 10-fold in parents and with integrity of more than 65% in mapping population individuals were selected for genetic mapping, and markers showing significant segregation distortion with chi-square test with *p* values < 0.01 were excluded. According to these parameters, only high-quality SNP markers were selected as potential

markers. The linkage relationship between two potential SNP markers was examined by the modified logarithm of odds (MLOD) scores. Linkage group assignments were under MLOD scores > 8.0 and a maximum recombination fraction of 0.4 [33]. The High-Map software and the SMOOTH algorithm were employed to order SNP markers and correct genotyping errors, respectively [34,35]. Genetic map distance was estimated using Kosambi's mapping function and indicated in centiMorgans (cM) [36]. The linkage map was evaluated by three steps: (1) the integrity of mapping markers in each individual in the mapping population, (2) haplotype mapping of each individual in all LGs, and (3) drawing the heat map of marker recombination relation.

## 2.7. Leaf Traits-Related QTLs Analysis

QTLs for leaf length-, leaf width-, leaf thickness-, and leaf number-related traits were detected using the software R/qtl [37]. Composite interval mapping (CIM) was used to map the leaf-related traits QTLs. The LOD threshold was used to evaluate the statistical significance of each QTL and was estimated using 1000 permutation tests with a confidence interval of 0.95. The location of each QTL was determined according to its LOD peak and the surrounding region. The proportion of variance explained (PVE) by a QTL peak was calculated using $1-10^{-2LOD/n}$, where n was the sample size [38]. Both the LOD score $\geq 3.0$ and PVE threshold $\geq 10\%$ were considered significant QTL intervals. Finally, the QTL profiles were visualized with MapChart 2.2 [39]. To identify candidate four leaf traits-related genes, SNP markers that were significantly correlated with four leaf-related traits were blasted to our transcriptome, which had been annotated to the publicly available databases (Nr, GO, KEGG, Swissprot, and COG), to explore the gene function of leaf-related traits-associated markers. Finally, these SNP markers corresponding unigenes were analyzed using GO enrichment online.

## 3. Results

### 3.1. Four Leaf Traits Data Evaluation of F1 Mapping Population

Leaf length (LL) and leaf width (LW) were quantitative traits, which were controlled by multiple genes. Data on LL and LW of 95 individuals of the F1 population and their parents were collected in 2018 (Figure 1, Tables 1 and S1). The two parents exhibited obvious differences in LL and LW (Table 1 and Figure S1). According to the frequencies analyses, the values of LL and LW of the F1 population varied continuously and belonged to a normal distribution, respectively (Figure 1 and Table 1). For leaf thickness (LT) (Table 1), the absolute value of skewness (1.318) slightly > 1, and of kurtosis (4.050) obviously < 10 demonstrated normal skewed distribution. For leaf number (LN) (Table 1), the absolute value of <1 for skewness and kurtosis, respectively, indicated the normal distribution of the F1 population data. For the four leaf-related traits, the CV ranged from 23.16% (LW) to 30.25% (LL) in the F1 population (Table 1). The $H^2$ values ranged from 36.02% (LN) to 96.48% (LL) in the F1 population (Table 1). The $H^2$ value of LN (36.02%) was much lower than that of LT (87.76%), while the $H^2$ value of LL (96.48%) was slightly higher than that of LW (96.13%) (Table 1).

The correlations between LL and the other three different leaf traits were significant: LL was positively correlated with LW, LT, and LN, respectively, with corresponding Spearman's correlation coefficients of 0.756, 0.480, and 0.268, respectively ($p < 0.01$) (Table 2). The LW was significantly and positively correlated with LT (0.397, $p < 0.01$), but not significantly correlated with LN ($p > 0.05$) (Table 2). The LT and LN showed no significant correlation with each other ($p > 0.05$).

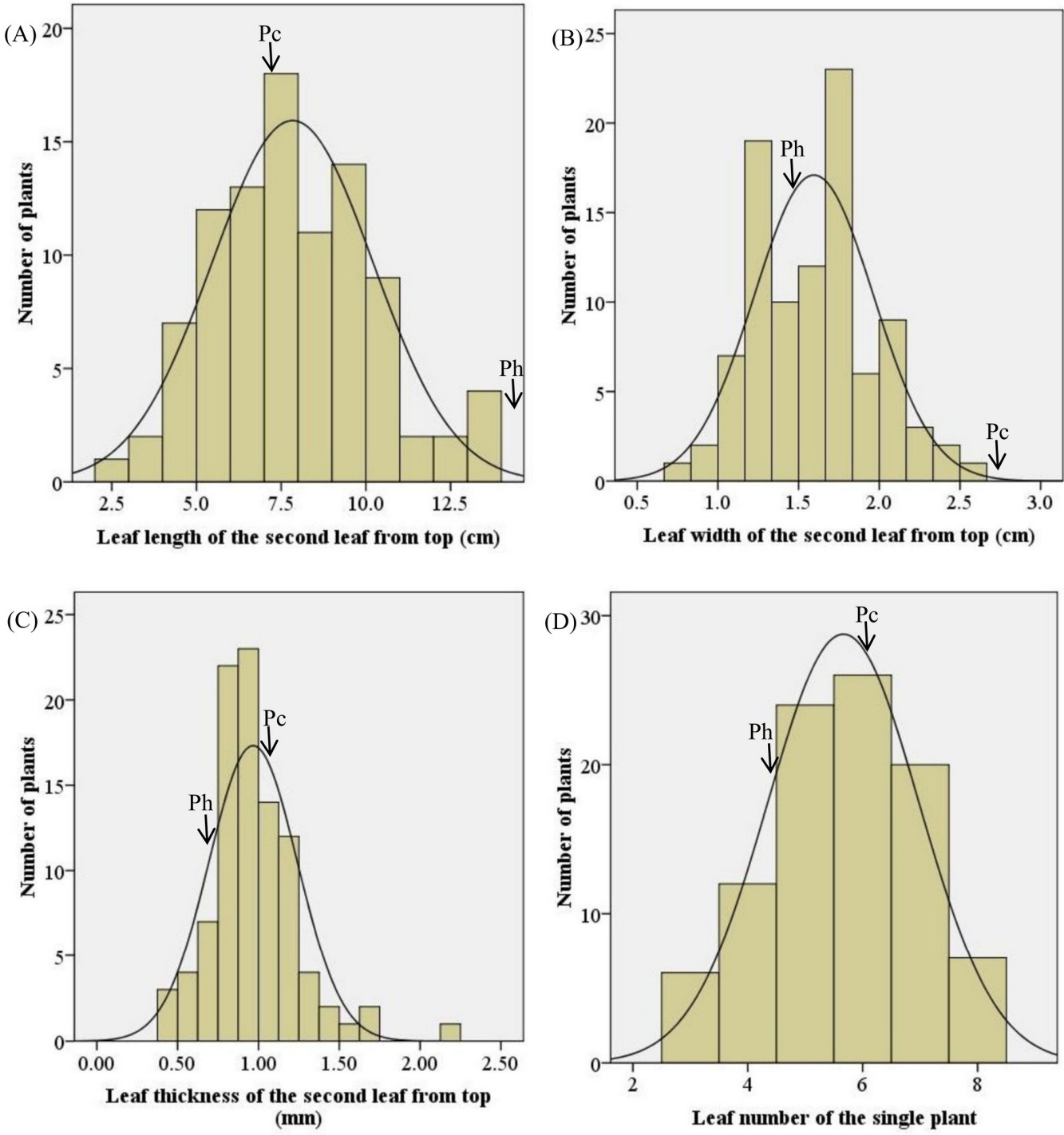

**Figure 1.** The distribution of four leaf-related traits including leaf length (**A**), leaf width (**B**), leaf thickness (**C**), and leaf number (**D**) in the F1 population. The *x*-axis shows the ranges of phenotypic traits and the *y*-axis represents the number of individuals in the F1 population. The values of the two parents *P. concolor* (Pc) and *P. hirsutissimum* (Ph) are indicated by arrows.

**Table 1.** Leaf traits statistics and estimate of heritability of the F1 population. Phenotipic values of both parents are also shown.

| Leaf Traits | Mean *P. concolor* | Mean *P. hirsutissimum* | F1 Mapping Population | | | | | | | | |
|---|---|---|---|---|---|---|---|---|---|---|---|
| | | | Max | Min | Mean $\pm$ se | SD | Variance | Skewness | Kurtosis | CV (%) | H$^2$ (%) |
| LL/cm | 7.48 | 17.66 | 13.70 | 2.20 | 7.84 $\pm$ 0.24 | 2.38 | 5.66 | 0.385 | 0.011 | 30.25 | 96.48 |
| LW/cm | 2.76 | 1.50 | 2.60 | 0.82 | 1.60 $\pm$ 0.04 | 0.37 | 0.14 | 0.242 | $-$0.392 | 23.16 | 96.13 |
| LT/mm | 1.05 | 0.69 | 2.20 | 0.47 | 0.97 $\pm$ 0.03 | 0.27 | 0.08 | 1.318 | 4.050 | 28.28 | 87.76 |
| LN/No. | 6 | 4.4 | 8 | 3 | 5.66 $\pm$ 0.14 | 1.32 | 1.74 | $-$0.178 | $-$0.574 | 23.28 | 36.02 |

Note: Max = maximum, Min = minimun, se = standard error, SD = standard deviation, LL = leaf length, LW = leaf width, LT = leaf thickness, LN = leaf number, CV = coefficient of variance, and H$^2$ = broad-sense heritability.

**Table 2.** Spearson's correlation coefficients between each of the four leaf traits in the F1 population of 95 individuals derived from *P. concolor* and *P. hirsutissimum*.

| Traits | LL | LW | LT | LN |
|---|---|---|---|---|
| LL | 1 | | | |
| LW | 0.756 ** | 1 | | |
| LT | 0.480 ** | 0.397 ** | 1 | |
| LN | 0.268 ** | 0.136 $^{NS}$ | 0.158 $^{NS}$ | 1 |

Note: LL = leaf length, LW = leaf width, LT = leaf thickness, and LN = leaf number. $^{NS}$ $p > 0.05$; ** $p < 0.01$.

### 3.2. RNA Sequencing Data and De Novo Assembly

RNA samples from 95 individuals of the F1 population and their two parents were extracted, respectively. The qualified RNA (Table S2) was used for cDNA library construction. After cDNA library construction and high-throughput sequencing, a total of 745.59 Gb clean data was generated. Among these clean reads, 91.38% achieved or exceeded a quality score of 30 (Q30, indicating a 99.9% confidence and 0.1% a chance of an error), and the GC (guanine-cytosine) content ranged from 46.59% to 51.40%. The clean data was 19.41 Gb for the female parent *P. concolor*, 18.95 Gb for the male parent *P. hirsutissimum*, and 4.15–10.46 Gb for the F1 progenies (Table S3). Trinity software was used for de novo assembly of the two parents' clean reads together into 24,866,677 contigs. With Illumina clean data information, the contigs were mapped back to the clean reads and can be detected from the same transcript and the distances between these contigs. Then, all these contigs were assembled into 349,730 transcripts and 185,089 unigenes, respectively (File S1). Among the unigenes, 22,291 unigenes were longer than 1000 bp. The average length and N50 length for all assembled unigenes were 545.36 bp and 788 bp, respectively (Table 3). All unigenes were aligned to public protein databases (Nr, Swiss-prot, GO, COG, and KEGG) by BLAST with E values $\leq 10^{-5}$. A total of 55,063 unigenes were annotated, accounting for 29.74% of all assembled unigenes (Table S4).

**Table 3.** Results of transcriptomes assembly from two *Paphiopedilum* parents and F1 offspring.

| Length Range | Contig | Transcript | Unigene |
|---|---|---|---|
| 200–300 bp | 24,760,054 (99.57%) | 127,279 (36.39%) | 95,156 (51.41%) |
| 300–500 bp | 53,461 (0.21%) | 72,952 (20.85%) | 42,698 (23.06%) |
| 500–1000 bp | 30,531 (0.12%) | 61,915 (17.70%) | 24,944 (13.47%) |
| 1000–2000 bp | 15,089 (0.06%) | 55,805 (15.95%) | 14,089 (7.61%) |
| >2000 bp | 7542 (0.03%) | 31,779 (9.087%) | 8202 (4.43%) |
| Total number | 24,866,677 | 349,730 | 185,089 |
| Total length | 1,124,466,789 | 276,064,986 | 100,941,049 |
| N50 length | 47 | 1430 | 788 |
| Mean length | 45.22 | 789.37 | 545.36 |

### 3.3. SNP Calling and Genotyping

The software GATK was used to identify parents and the F1 progenies' SNP markers combined with the assembled unigenes. In total, 504,936 SNP markers were developed,

in which 165,196 polymorphic SNPs were successfully encoded and grouped into eight segregation patterns (ab × cd, ef × eg, ab × cc, cc × ab, hk × hk, lm × ll, nn × np, and aa × bb) (Table 4 and Figure 2). Among these encoded polymorphic SNPs, biallelic and tri-allelic SNPs had rates of 98.20% and 1.80%, respectively (Table 4). The nature of biallelic SNPs was also investigated (Table 4). Most were transition-type SNPs of R (A/G) and Y (C/T) types with rates of 30.40% and 30.61% of all encoded SNPs, respectively. Four transversion type SNPs were identified, including W(A/T), M(A/C), K(G/T), and S(C/G) with frequencies ranging from 7.87% to 12.16% of all encoded SNPs. Since the F1 population was heterozygous, the markers from the segregation pattern of aa x bb were filtered out. A total of 66,904 SNPs from the other seven segregation patterns were used for map construction. To further enhance the genetic map accuracy, only SNPs with average sequence depths ≥ 10-fold in parents, greater than 65% coverage degree, and $0.01 < p$ value of segregation distortion $< 0.05$ were used for map construction. Finally, 9029 of the 66,904 SNP markers were screened out for genetic map construction.

**Table 4.** Statistic of identified encoded SNP markers types.

| Type | Number | Ratio |
|---|---|---|
| Total number of encoded SNPs | 165,196 | 100% |
| Biallelic SNPs | 162,219 | 98.20% |
| W(A/T) | 20,097 | 12.16% |
| R(A/G) | 50,225 | 30.40% |
| M(A/C) | 14,249 | 8.62% |
| K(G/T) | 14,059 | 8.51% |
| Y(C/T) | 50,573 | 30.61% |
| S(C/G) | 13,016 | 7.87% |
| ≥Tri-allelic SNPs | 2977 | 1.80% |

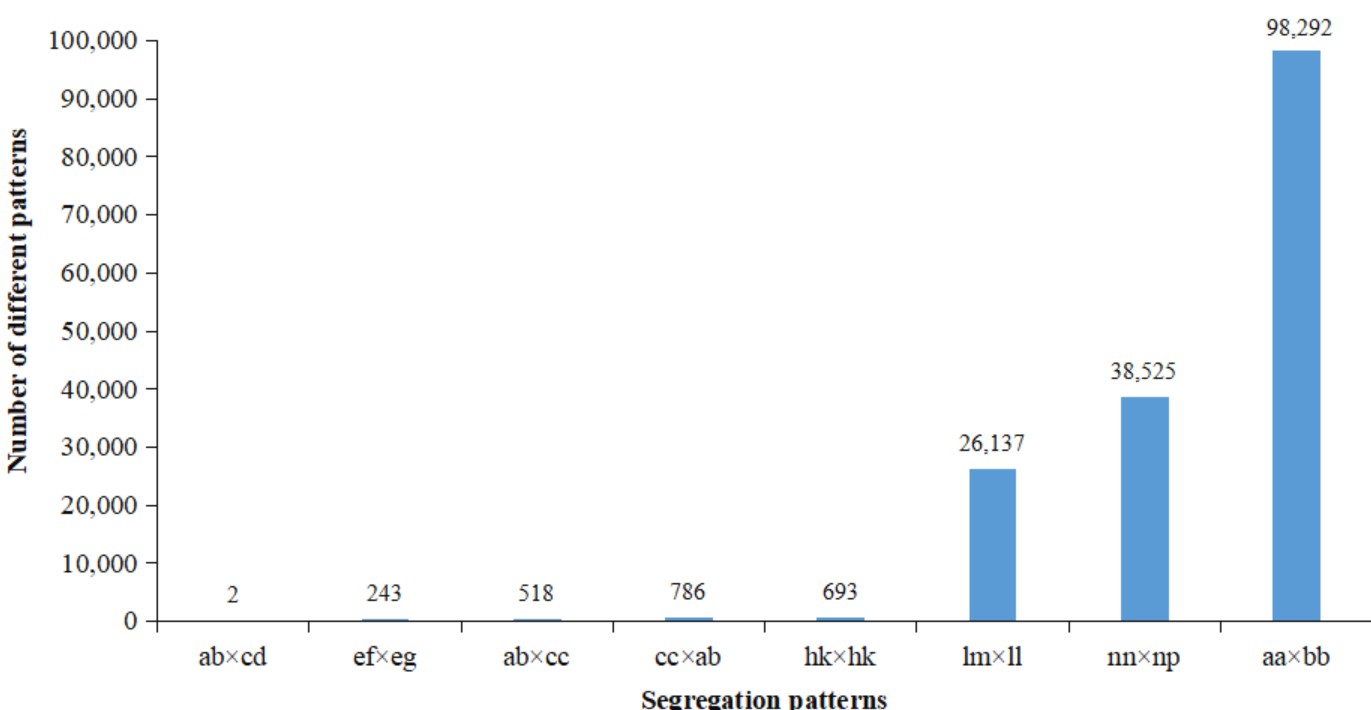

**Figure 2.** Number of polymorphic SNP markers in each of the eight segregation patterns in *Paphiopedilum*.

*3.4. High-Density Genetic Linkage Map*

After a series of screenings, the identified 9029 SNPs were used to construct the genetic map for slipper orchids. Then, according to the linkage analysis, 8410 SNPs were found to be

effective and used for the final map construction (Table S5). HighMap software assigned all of the 8410 SNPs to 13 linkage groups (LG) (Figure 3 and Table S6). The integrated genetic map spanned 1616.18 cM, with an average distance of 0.19 cM between adjacent markers (Table 5). LG1 contained the fewest markers of 183 and spanned a length of 132.98 cM, with an average distance of 0.73 cM and a maximum gap of 9.17 cM being observed between adjacent markers. LG9 harbored the largest number of markers of 1468 and covered a length of 114.35 cM, with an average distance of only 0.08 cM and a maximum gap of only 5.98 cM being observed between adjacent markers. The longest LG was LG4 (189.89 cM), which had 321 markers, while the shortest LG was LG10 (99.52 cM), which possessed 714 markers. On average, each LG comprised 646 markers. The "Gap < 5 cM" value of the 13 LGs ranged from 97.25% to 100.00%, with an average value of 99.24%.

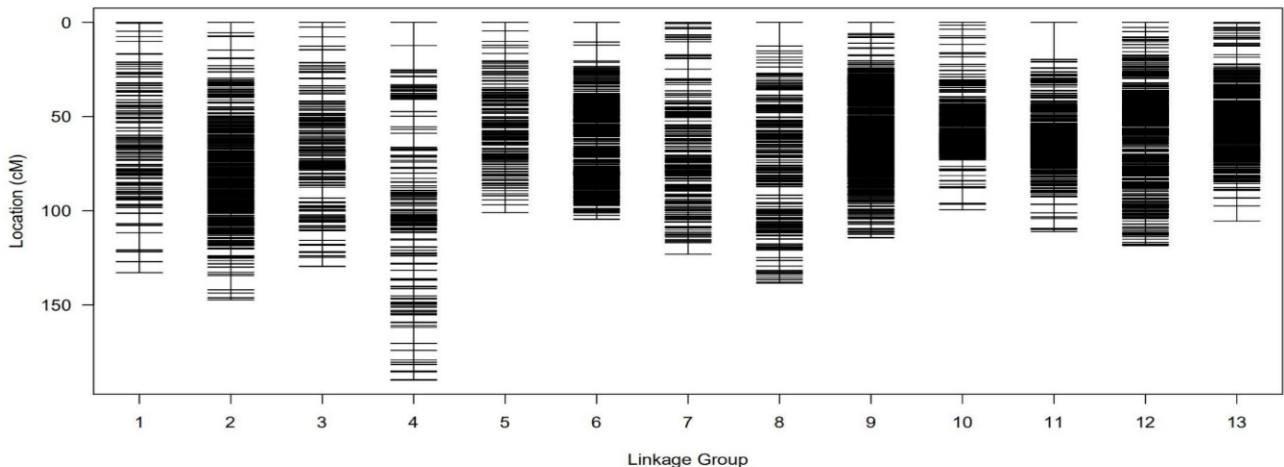

**Figure 3.** Distribution of SNP markers on the 13 linkage groups of *Paphiopedilum*.

The female and male maps were also constructed in the current study. The total number of markers used for female and male maps were 5498 and 3173, respectively (Tables 5, S7 and S8, Figures S2 and S3). The total genetic lengths of female and male maps were 1994.90 cM and 1670.55 cM, respectively, with average distances between adjacent markers in the female and male maps of 0.36 cM and 0.53 cM, respectively (Table 5). Some significant differences were also observed in some LGs, for example, the markers of LG1, LG4, LG6, LG10, and LG12 in the female map were much less than those in the male map. However, the markers of LG3, LG5, LG7, LG8, LG9, LG11, and LG13 in the female map were much more than those in the male map.

According to the chi-square test of the 8410 SNPs, 1877 SNPs exhibiting segregation distortion with p values < 0.05 were retained even though the extremely significant SNPs with p values < 0.01 were excluded (Table 5). The total segregation distortion ratio was 22.31%. The most segregation distortion SNPs were observed for LG9 (798 SNPs), followed by LG6 (314 SNPs), LG8 (207 SNPs), and LG4 (189 SNPs). The markers on these four LGs were significantly improved after decreasing coverage degree and increasing p value. However, the four linkage groups had a notably lower number of markers if not selected for segregation distortion. The rest of the 9 LGs had segregation distortion markers below 95. Among them, the LG3 contained the least segregation distortion markers, only 3 SNPs.

**Table 5.** Description of the basic characteristics of the 13 linkage groups in *Paphiopedilum* orchids.

| LGs | Total Markers | | | Total Distance (cM) | | | Average Distance (cM) | | | Gap < 5 cM (%) | | | Max Gap (cM) | | | No. of SD [a] |
|-----|---------------|---|---|---------------------|---|---|------------------------|---|---|----------------|---|---|--------------|---|---|------------|
| | Female Map | Male Map | Integrated Map | Female Map | Male Map | Integrated Map | Female Map | Male Map | Integrated Map | Female Map | Male Map | Integrated Map | Female Map | Male Map | Integrated Map | |
| LG1 | 3 | 183 | 183 | 511.45 | 132.98 | 132.98 | 170.48 | 0.73 | 0.73 | 0.00 | 97.25 | 97.25 | 425.86 | 9.17 | 9.17 | 32 |
| LG2 | 38 | 333 | 688 | 134.61 | 147.06 | 147.33 | 0.35 | 0.44 | 0.21 | 99.74 | 99.10 | 99.56 | 10.95 | 14.59 | 7.41 | 26 |
| LG3 | 223 | 0 | 223 | 129.74 | 0.00 | 129.74 | 0.58 | 0.00 | 0.58 | 98.20 | 0.00 | 98.20 | 6.66 | 0.00 | 6.66 | 3 |
| LG4 | 7 | 320 | 21 | 183.82 | 189.90 | 189.89 | 26.26 | 0.59 | 0.59 | 0.00 | 97.81 | 97.81 | 63.69 | 12.91 | 12.91 | 189 |
| LG5 | 254 | 6 | 254 | 101.04 | 76.10 | 101.04 | 0.4 | 12.68 | 0.40 | 99.60 | 20.00 | 99.60 | 5.41 | 25.30 | 5.41 | 8 |
| LG6 | 37 | 747 | 767 | 129.41 | 104.70 | 104.58 | 3.5 | 0.14 | 0.14 | 69.44 | 99.73 | 99.74 | 21.49 | 10.58 | 10.44 | 314 |
| LG7 | 310 | 10 | 316 | 117.06 | 467.35 | 123.08 | 0.38 | 46.74 | 0.39 | 99.03 | 22.22 | 98.73 | 6.93 | 260.04 | 6.92 | 95 |
| LG8 | 414 | 7 | 414 | 138.42 | 52.24 | 138.42 | 0.33 | 7.46 | 0.33 | 99.76 | 66.67 | 99.76 | 12.60 | 21.10 | 12.60 | 207 |
| LG9 | 1466 | 30 | 1468 | 114.35 | 107.74 | 114.35 | 0.08 | 3.59 | 0.08 | 99.93 | 72.41 | 99.93 | 5.98 | 13.66 | 5.98 | 798 |
| LG10 | 31 | 709 | 714 | 95.71 | 99.52 | 99.52 | 3.09 | 0.14 | 0.14 | 86.67 | 99.72 | 99.86 | 19.70 | 8.15 | 8.15 | 29 |
| LG11 | 1080 | 43 | 1085 | 111.10 | 100.32 | 111.10 | 0.1 | 2.33 | 0.10 | 99.81 | 88.10 | 99.82 | 19.59 | 18.98 | 19.59 | 65 |
| LG12 | 87 | 706 | 754 | 122.61 | 118.63 | 118.57 | 1.41 | 0.17 | 0.16 | 91.86 | 100.00 | 100.00 | 11.23 | 3.06 | 2.84 | 42 |
| LG13 | 1204 | 79 | 1223 | 105.58 | 74.01 | 105.58 | 0.09 | 0.94 | 0.09 | 99.92 | 98.72 | 99.92 | 12.26 | 5.45 | 8.11 | 69 |
| Total | 5498 | 3173 | 8410 | 1994.90 | 1670.55 | 1616.18 | 0.36 | 0.53 | 0.19 | 80.30 | 80.14 | 99.24 | 425.86 | 260.04 | 19.59 | 1877 |

Note: SD [a] represents the SNP markers with segregation distortion of *p* values < 0.05 in integrated map.

### 3.5. Quality Evaluation of the Genetic Map

To evaluate the quality of the genetic map, integrity distribution, haplotype mapping, and heat mapping were carried out. The high integrity reflected the accuracy of the genotyping of the mapping population. In this study, the average integrity of each individual in the F1 mapping population was 99.71% (Figure S4). These results guaranteed the accuracy of genotyping in the F1 population. The haplotype map indicated the possible double exchange of the population, which was caused by a genotyping error or a possible recombination hotspot. The haplotype maps of each LGs were developed using 8410 SNP markers. The results showed that the origin of larger segments in each individual kept consistent, which demonstrated the high quality of the genetic map (File S2). The heat maps were drawn based on the pair-wise recombination value from the 8410 mapped SNP markers to indicate the recombination relationship between mapped markers on each LG. The results exhibited that the linkage relationships between adjacent markers on each LG were very strong, and the linkage relationships between distant markers were gradually weakened, indicating the correct order of mapped SNP markers (File S3).

### 3.6. Identifying QTLs for Four Leaf-Related Traits in Slipper Orchids

Using R/qtl with our constructed high-density SNP linkage map, a total of 12 QTLs associated with four leaf-related traits were detected in the F1 mapping population. They were distributed on 8 LGs, including 3 LL QTLs, 3 LW QTLs, 3 LT QTLs, and 3 LN QTLs with LOD scores ranging from 3.053 to 5.014 (Figure 4, Tables 6 and S9). Both LG2 and LG9 had the highest QTLs, three for each LG, while LG1, LG3, LG5, LG7, LG10, and LG11 each only contained one QTL.

**Table 6.** Statistics of QTLs for four leaf-related traits detected in the F1 population of *Paphiopedilum* orchids.

| Leaf Traits | QTLs | Linkage Group | Linkage Map Position Start (cM) | Linkage Map Position Final (cM) | Interval Size (cM) | Max LOD | PVE (%) [a] | SNP No. |
|---|---|---|---|---|---|---|---|---|
| LL | qLL2-1 | 2 | 73.375 | 73.375 | 0.00 | 3.099 | 11.86 | 1 |
| LL | qLL9-1 | 9 | 7.986 | 23.413 | 15.427 | 3.313 | 12.41–12.65 | 2 |
| LL | qLL10-1 | 10 | 64.342 | 64.645 | 0.303 | 4.472 | 13.86–19.49 | 3 |
| LW | qLW1-1 | 1 | 127.059 | 132.982 | 5.923 | 4.087 | 14.33–17.97 | 10 |
| LW | qLW7-1 | 7 | 102.095 | 102.095 | 0.00 | 3.121 | 14.04 | 2 |
| LW | qLW9-1 | 9 | 50.358 | 91.793 | 41.435 | 5.014 | 16.88–21.58 | 5 |
| LT | qLT2-1 | 2 | 73.483 | 73.483 | 0.00 | 3.317 | 14.85 | 2 |
| LT | qLT5-1 | 5 | 87.770 | 87.770 | 0.00 | 4.948 | 21.33 | 2 |
| LT | qLT9-1 | 9 | 24.727 | 81.723 | 56.996 | 4.106 | 15.59–18.05 | 4 |
| LN | qLN2-1 | 2 | 61.984 | 61.984 | 0.00 | 3.411 | 15.24 | 2 |
| LN | qLN3-1 | 3 | 56.101 | 56.101 | 0.00 | 3.142 | 12.34 | 1 |
| LN | qLN11-1 | 11 | 52.537 | 52.537 | 0.00 | 3.228 | 14.49 | 4 |

Note: LL = leaf length, LW = leaf width, LT = leaf thickness, and LN = leaf number. [a] Indicates the percentage of phenotypic variation explained.

Three QTL loci, designated qLL2-1, qLL9-1, and qLL10-1, were detected for LL and were located on LG2, LG9, and LG10, respectively (Table 6 and Figure 4A). The most prominent QTL, qLL10-1, with phenotypic variance explained (PVE) values ranging from 13.86% to 19.49%, followed by qLL9-1, which explained 12.41% to 12.65% of the leaf length variations, respectively. The lowest contribution rate was 11.86% for eqLL2-1 (Tables 6 and S9). Among the three SNP markers corresponding to qLL10-1, Marker77321 and Marker122309 were at the start position (64.342 cM) and the final position (64.645 cM), respectively, whereas the other Marker103037 covered the interval of 64.342–64.645 cM.

For LW, 3 QTLs were located in LG1 (qLW1-1), LG7 (qLW7-1), and LG9 (qLW9-1) with PVE values ranging from 14.04% to 21.58% (Figure 4B, Tables 6 and S9). Interestingly, the final position of qLW9-1 located at 91.793 cM along LG9, accounted for the highest contribution rate of 21.58% (Table 6). Among the qLW9-1 region, three SNP markers (Marker83301, Marker155132, and Marker126719) were located at the same start position of 50.358 cM, whereas the other two SNP markers (Marker80443 and Marker80444) were located at the same final position of 91.793 cM. The qLW1-1 could explain 14.33% to 17.97% of the leaf width variations, and ten SNP markers were uncovered within this QTL region.

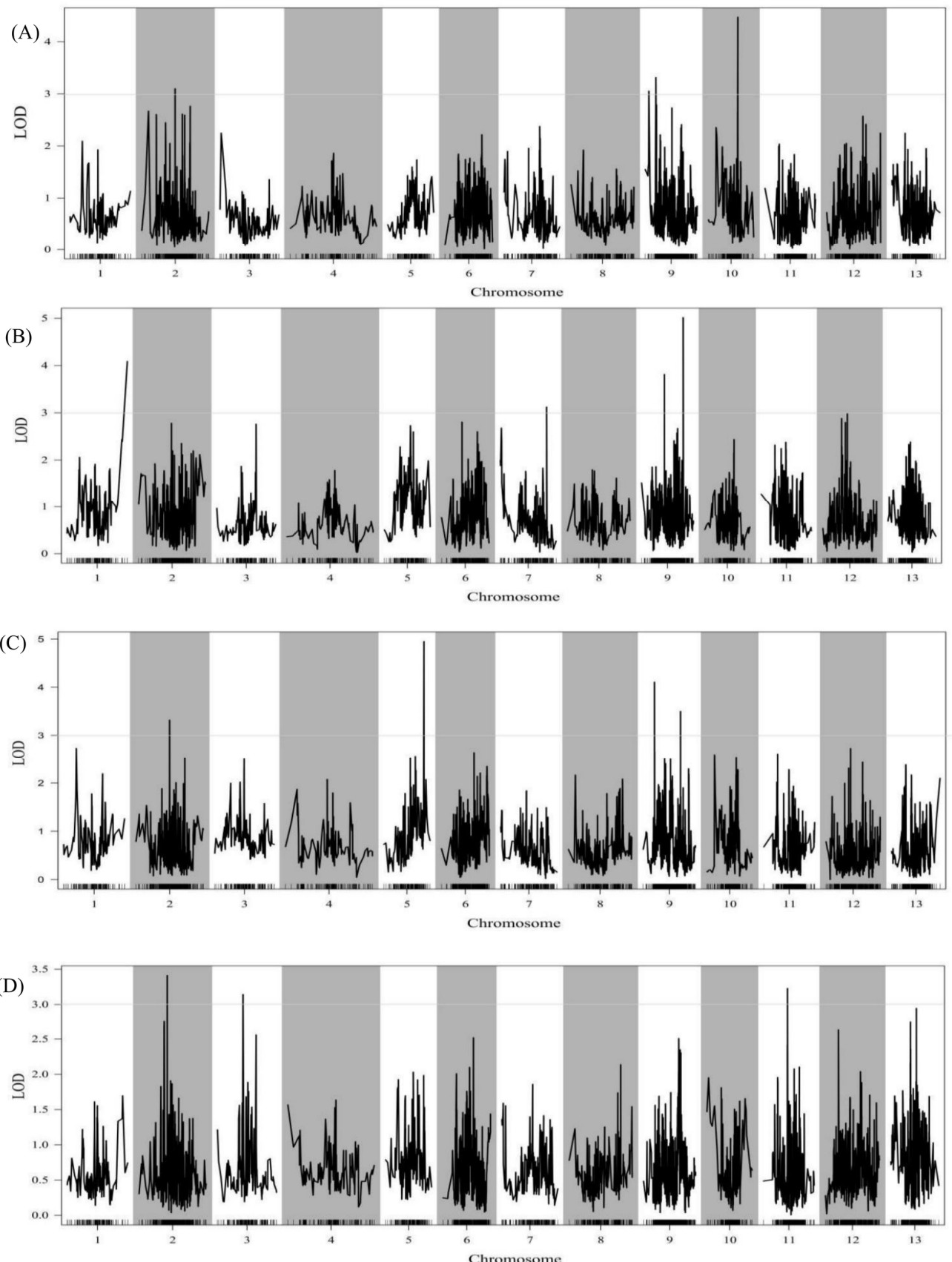

**Figure 4.** QTLs analysis of four leaf-related traits using the high-density genetic map of *Paphiopedilum*. (**A**) leaf length, (**B**) leaf width, (**C**) leaf thickness, and (**D**) leaf number.

Three QTLs were detected on 3 LGs (LG2, LG5, and LG9) for the LT, one for each LG (Table 6 and Figure 4C). The highest contribution rate was 21.33% for qLT5-1 on LG5, followed by qLT9-1 (15.59–18.05%) on LG9 and qLT2-1 (14.85%) on LG2. For qLT5-1, both two SNP markers (Marker151869 and Marker151866) within this region were positioned at 87.770 cM with an LOD score of 4.948. Among four SNP markers corresponding to qLT9-1, three markers (Marker121454, Marker121464, and Marker121457) were at the same position of 24.727 cM, whereas the other one (Marker17971) was located at 81.723 cM. The case in qLT2-1 was observed for two SNP markers, Marker101111 and Marker93699, which were at the same position of 73.483 cM (Table S9).

Three QTLs were detected for the LN, qLN2-1, qLN3-1, and qLN11-1 on LG2, LG3, and LG11, respectively (Figure 4D, Tables 6 and S9). The qLN2-1 had a prominent effect on the slipper orchids' leaf number, explaining 15.24% of the phenotypic variation with an LOD score of 3.411. Two SNP markers (Marker49462 and Marker62874) were located at the position of 61.984 cM on LG2 in this QTL. For qLN11-1, four SNP markers (Marker13507, Marker13508, Marker13509, and Marker13510) were located at the same position of 52.537 cM on LG11, each marker with a contribution rate of 14.49%, respectively. For qLN3-1, one SNP marker (Marker157165) was located at the position of 56.101 cM on LG3, with the lowest contribution rate of 12.34%. Especially, LL, LW, and LT-related QTLs were all found located on LG9 (Figure 4 and Table 6).

### 3.7. Potential Candidate Genes in Four Leaf Traits-Related QTLs

There was a total of 38 SNP markers in the regions of the 4 leaf traits-associated QTLs (Table 6). Among these QTLs, both qLL2-1 and qLN3-1 were single SNP marker-associated QTLs, while qLW1-1 had the largest SNP marker number with 10. Based on the public database's annotations of our assembled transcriptome and our constructed genetic map, we determined that 38 SNPs in 12 leaf traits-associated QTL regions were located in 25 unigenes (Table S10). Nine of these 25 unigenes were mapped to public KEGG pathways, they encoded aminoacylase (K14677; c188082.graph_c0), translation initiation factor 4G (K03260; c193404.graph_c0), chloroplastic DNA-directed RNA polymerase 3B (K10908; c193047.graph_c1), peroxisome biogenesis factor 10 (K13346; c182003.graph_c0), chloroplastic omega-amidase (K13566; c181454.graph_c1), UDP-arabinose 4-epimerase (K12448; c186860.graph_c1), methyltransferase-like protein 6 (K00599; c186538.graph_c0), isocitrate dehydrogenase (K00031; c173310.graph_c0), and chloroplastic 50S ribosomal protein L24 (K02895; c170791.graph_c0).

GO enrichment analysis indicated that 11 of these 25 unigenes were enriched with a total number of 59 GO terms in various biological processes, cellular components, and molecular functions (Tables 7 and S11). The unigene c190842.graph_c0 in qLL10-1 has 12 GO terms of biological processes: embryo development ending in seed dormancy (GO:0009793), rRNA processing (GO:0006364), thylakoid membrane organization (GO:0010027), chloroplast organization (GO:0009658), GTP catabolic process (GO:0006184), response to light stimulus (GO:0009416), and mRNA modification (GO:0016556); cellular component: chloroplast inner membrane (GO:0009706) and chloroplast stroma (GO:0009570); and molecular function: magnesium ion binding (GO:0000287), GTP binding (GO:0005525), and GTPase activity (GO:0003924). The unigene c189030.graph_c0 in qLL10-1 has 11 GO terms of biological processes: DNA recombination (GO:0006310), protein modification by small protein conjugation or removal (GO:0070647), DNA replication (GO:0006260), meiosis I (GO:0007127), regulation of transcription, DNA-templated (GO:0006355), anatomical structure development (GO:0048856), macromolecule methylation (GO:0043414), negative regulation of gene expression (GO:0010629), response to stress (GO:0006950), histone modification (GO:0016570), and post-embryonic development (GO:0009791). The unigene c188082.graph_c0 in qLT2-1 has 10 GO terms of biological processes: RNA splicing, via endonucleolytic cleavage and ligation (GO:0000394), response to zinc ion (GO:0010043), methionine biosynthetic process (GO:0009086), proteolysis (GO:0006508), and cysteine biosynthetic process (GO:0019344); cellular component: endo-

plasmic reticulum (GO:0005783), Golgi apparatus (GO:0005794), and vacuole (GO:0005773); and molecular function: aminoacylase activity (GO:0004046) and metallopeptidase activity (GO:0008237). In addition, the unigene c182003.graph_c0 in qLW7-1 has 8 GO terms of biological processes: photorespiration (GO:0009853), fatty acid beta-oxidation (GO:0006635), attachment of peroxisome to chloroplast (GO:0010381), embryo development ending in seed dormancy (GO:0009793), and protein import into peroxisome matrix (GO:0016558); cellular component: cytosol (GO:0005829) and integral component of peroxisomal membrane (GO:0005779); and molecular function: zinc ion binding (GO:0008270). Finally, the remaining 7 unigenes were enriched in 1 to 3 GO terms, respectively, mainly involved in translation (GO:0006412), ribosome (GO:0005840), structural constituent of ribosome (GO:0003735), membrane (GO:0016020), methylation (GO:0032259), galactose metabolic process (GO:0006012), UDP-glucose 4-epimerase activity (GO:0003978), plasma membrane (GO:0005886), and cellular metabolic process (GO:0044237) (Table S11).

**Table 7.** Information on the candidate SNP markers and their corresponding unigenes with GO enrichment.

| SNP Marker Name | Corresponding Unigene Name | SNP Site in Unigene (bp) | LGs | Position in LG (cM) | Description (Nr Database) |
|---|---|---|---|---|---|
| Marker62778 | c184170.graph_c0 | 352 | 1 | 132.982 | CSC1-like protein RXW8 |
| Marker62785 | c184170.graph_c0 | 937 | 1 | 132.982 | CSC1-like protein RXW8 |
| Marker62795 | c184170.graph_c0 | 1740 | 1 | 132.982 | CSC1-like protein RXW8 |
| Marker62797 | c184170.graph_c0 | 1804 | 1 | 132.982 | CSC1-like protein RXW8 |
| Marker62892 | c184183.graph_c0 | 2000 | 1 | 127.059 | Leucine carboxyl methyltransferase |
| Marker93699 | c188082.graph_c0 | 982 | 2 | 73.483 | Aminoacylase-1 |
| Marker101111 | c188843.graph_c0 | 1217 | 2 | 73.483 | Plant intracellular Ras-group-related LRR protein 4 |
| Marker151866 | c193047.graph_c1 | 1021 | 5 | 87.77 | DNA-directed RNA polymerase 3, chloroplastic |
| Marker151869 | c193047.graph_c1 | 1871 | 5 | 87.77 | DNA-directed RNA polymerase 3, chloroplastic |
| Marker49769 | c182003.graph_c0 | 1033 | 7 | 102.095 | peroxisome biogenesis factor 10 |
| Marker83301 | c186860.graph_c1 | 695 | 9 | 50.358 | Probable UDP-arabinose 4-epimerase 3 |
| Marker126719 | c191195.graph_c0 | 1338 | 9 | 50.358 | Pentatricopeptide repeat domain |
| Marker103037 | c189030.graph_c0 | 864 | 10 | 64.588 | uncharacterized protein LOC103703380 isoform X1 |
| Marker122309 | c190842.graph_c0 | 1789 | 10 | 64.645 | Chloroplastic GTP-binding protein ObgC1 |
| Marker13507 | c170791.graph_c0 | 471 | 11 | 52.537 | Chloroplastic 50S ribosomal protein L24 |
| Marker13508 | c170791.graph_c0 | 492 | 11 | 52.537 | Chloroplastic 50S ribosomal protein L24 |
| Marker13509 | c170791.graph_c0 | 585 | 11 | 52.537 | Chloroplastic 50S ribosomal protein L24 |
| Marker13510 | c170791.graph_c0 | 684 | 11 | 52.537 | Chloroplastic 50S ribosomal protein L24 |

## 4. Discussion

### 4.1. High-Density SNP Genetic Map for Lady's Slipper Orchids

Molecular marker development and genetic map construction are two important basic works for carrying out molecular breeding in orchids [19–21,24,26]. Previous studies used traditional markers including EST-SSR, ISSR, SRAP and RAPD, and newly developed SNP markers to construct genetic maps in *Dendrobium* orchids [19–21,24,26]; however, large SNP markers' development in lady's slipper orchids lagged behind *Dendrobium* species due to their relatively large and complex genome [7]. Our preliminary experiments discovered that SLAF sequencing was not an ideal approach for developing SNP markers due to their low polymorphism rate 3.0–5.3% in *P. concolor* and *P. hirsutissimum*, which was lower than the polymorphism rate of 5.89% in *D. nobile* and *D. wardianum* [26]. Therefore, it is almost impossible to construct a high-density genetic map in *P. concolor* and *P. hirsutissimum* by SLAF sequencing.

RNA sequencing is a high-throughput technique that can efficiently develop large numbers of SNP markers in a short time to generate enough polymorphic markers for high-density genetic map construction [23,26,40]. Our study has reported the densest genetic map for lady's slipper orchids (*P. concolor* and *P. hirsutissimum*) published so far. The present integrated map contained 8410 SNP markers distributed on 13 LGs, and covered

1616.18 cM with an average distance of 0.19 cM between adjacent markers and an average of 646 markers per LG. Compared with an average map distance of 0.76 cM between adjacent markers in *Chrysanthemum* [25], an average distance of 0.32 cM between adjacent markers in *D. moniliforme* and *D. officinale* [24], and an average distance of 0.41 cM between adjacent markers in *D. nobile* [26], the genetic map constructed in this study had a lower average map distance between adjacent markers. Meanwhile, in comparison to 239 markers per LG in *Chrysanthemum* [25], 451 markers per LG in *D. moniliforme* and *D. officinale* [24], and 503 markers per LG in *D. nobile* [26], the current study had higher average markers per LG. Therefore, our study confirmed that RNA sequencing was suitable for constructing a high-density genetic map in lady's slipper orchids. The high-density genetic map studied here will be a useful platform for the assembly of *P. concolor* and *P. hirsutissimum* genomes, as well as molecular marker-assisted selection breeding, map-based cloning, and comparative genomes analyses.

### 4.2. QTLs Identified for Leaf-Related Traits in Lady's Slipper Orchids

The genetic controlling leaf traits in lady's slipper orchids remains poorly researched, in contrast to the situation in model *Arabidopsis* [9], oilseed rape [11,41], soybean [15], common bean [16], poplar [17,42], *Catalpa bungei* [43], and Chinese bayberry [44]. The two parents used in the present study, *P. concolor* (female) and *P. hirsutissimum* (male), have contrasting leaf phenotypes [3,6], making them valuable for investigating the molecular mechanisms of leaf-related traits. Using the high-density SNP map and the F1 mapping population, we identified a total of 12 QTLs for four leaf-related traits (Figure 4 and Table 6). While no QTL loci were detected on five LGs (4, 6, 8, 12, and 13), all of the other eight LGs had QTLs distributions.

Leaf length and leaf width-related QTLs have also been identified in many reports. In an F1 poplar population, two QTLs were identified on LG16: 96.0–137.2 and LG1: 40.0–46.0 in spring, respectively, which explained 4.66% and 7.15% of the phenotypic variations for leaf length and leaf width, respectively [17]. In the F1 population of *Catalpa bungei* × *Catalpa duclouxii*, one QTL (Q16–60) was mapped to LG16, explaining 16.51% of the phenotypic variation in leaf length with an LOD score of 17.64 [43]. In an F1 Chinese bayberry, three leaf length-related QTLs were located in LG3 (148.68–154.38 cM), LG5 (165.21–174.59 cM), and LG6 (132.82–133.27 cM), with LOD values of 2.58, 3.06, and 2.64, explaining 7.82%, 9.42%, and 7.78% of the observed genotypic variation, respectively [44]. In the present study, three leaf length-related QTLs were identified on LG2: 73.375 cM (qLL2-1), LG9: 7.986–23.413 cM (qLL9-1), and LG10: 64.342–64.645 cM (qLL10-1), respectively, which explained 11.86%, 12.41–12.65% and 13.86–19.49% of the phenotypic variation for leaf length, respectively. Concerning leaf width, we found that three QTLs were located on LG9 (qLW9-1, 50.358–91.793 cM), LG1 (qLW1-1, 127.059–132.982 cM), and LG7 (qLW7-1, 102.095 cM), with contribution rates ranging from 16.88–21.58%, 14.33–17.97%, and 14.04%, respectively (Table 6). Although we only detected the leaf phenotypic data of the F1 population in 2018, the correlation between leaf length and leaf width in the F1 population was high (0.756). Moreover, high $H^2$ values were observed for LL, LW, and LT in the F1 population (Table 1). These results suggested that the variations of LL, LW, and LT in the F1 population were mainly caused by heredity and small effects of environmental variation. As we know, *Paphiopedilum* plants have long juvenile periods of about 3–6 years and slow growth characteristics. Therefore, the leaf length, leaf width and leaf thickness-related QTLs identified here can be considered stable.

In addition, we found that one QTL for leaf thickness (qLT9-1) was positioned at 24.727–81.723 cM on LG9 (Table 6). Interestingly, the end (23.413 cM) of qLL9-1 for leaf length and the start (24.727 cM) of qLT9-1 for leaf thickness were very close. Moreover, the end (81.723 cM) of qLT9-1 for leaf thickness was positioned in qLW9-1 (50.358–91.793 cM) for leaf width. Meanwhile, we also found that qLL2-1 (73.375 cM) for leaf length and qLT2-1 (73.483 cM) for leaf thickness were very close on LG2, and that qLN2-1 (61.984 cM) for leaf number was also located on LG2. These results suggest that these six QTLs on LG9

and LG2 may play important roles in controlling the formation of leaf length, leaf width, leaf thickness, and leaf number in *Paphiopedilum* plants. The information generated in this study is the first report of QTL mapping for leaf length, leaf width, leaf thickness, and leaf number in slipper orchids.

### 4.3. Candidate Genes Associated with Leaf-Related Traits in Slipper Orchids

Based on QTLs and functional annotation, 25 candidate unigenes were screened and identified for four leaf-related traits, of which 11 unigenes were enriched with GO terms (Tables 7 and S11). The unigene c190842.graph_c0 in qLL10-1 with most GO enrichment terms (12), was predicted to encode chloroplastic GTP-binding protein 1 (ObgC1). ObgC1 was conserved in most organisms, from bacteria to eukaryotes, which had diverse and important functions in bacteria, including morphological development, DNA replication, and ribosome maturation [45,46]. In *Arabidopsis*, a plant ortholog of bacterial *Obg*, *AtObgC* was essential for early embryogenesis [45]. In *Oryza sativa*, *ObgC1* acted as a key regulator of DNA replication and ribosome biogenesis in chloroplast nucleoids [46]. In epiphytic plant *D. officinale*, *ObgC* had an evolutionarily conserved role in ribosome biogenesis to adapt chloroplast development to the environment [47]. In a previous report, leaf transcriptomes differences analysis and leaf internal morphology observation between *P. concolor* and *P. hirsutissimum* demonstrated that chloroplast-related genes probably played crucial roles in leaf formation [6]. Thus, we assume that the encoded protein ObgC1 of the Marker122309/c190842.graph_c0 may be involved in ribosome biogenesis in chloroplast and leaf length development to adapt to the environment, and this marker/unigene is worthy of further study.

Of five SNPs in qLW9-1, two SNP markers annotated genes, c186860.graph_c1 and c191195.graph_c0, were encoded probable UDP-arabinose 4-epimerase 3 and pentatricopeptide repeat domain protein (PPR), respectively. *UDP-arabinose 4-epimerase 3*/c186860.graph_c1 was also differentially expressed between *P. concolor* and *P. hirsutissimum* leaves transcriptomes [6]. This gene was coordinated with the incorporation of pentose sugars onto cell walls in barley leaves [48]. PPR proteins were widely found in plants and played various functions in organellar metabolism, for example, PPR647 was crucial for RNA editing and RNA splicing of chloroplast genes, and played an essential role in chloroplast development in maize [49]. One other SNP in qLW7-1, namely, Marker49769, its annotated gene c182003.graph_c0, was with 10 GO enrichment terms, encoding peroxisome biogenesis factor 10 (PEX10). PEX10 was essential for the maintenance of endoplasmic reticulum morphology and contributed to the biosynthesis of cuticular wax in *Arabidopsis* [50]. These results suggest that c186860.graph_c1, c191195.graph_c0, and c182003.graph_c0 may be involved in sugar transportation onto cell walls, chloroplast development, endoplasmic reticulum morphology, and biosynthesis of cuticular wax during the leaf width formation process in slipper orchids, respectively. However, further study is warranted to detect the exact function of these three candidate genes in the formation of leaf width in slipper orchids.

Another gene is of particular interest in qLT2-1, namely, c188082.graph_c0, which encodes aminoacylase-1 (ACY-1). ACY-1 was a zinc-binding enzyme that was important in urea cycling, ammonia scavenging, and oxidative stress responses in animals; in plants, overexpression of *ZmACY-1 of Zea mays* in *Nicotiana benthamiana* promoted growth rate by promoting growth-related genes [51]. The other gene in qLN11-1, c170791.graph_c0, is predicted to encode large subunit ribosomal protein L24 (RPL24). *RPL24* was implicated in translation reinitiation of polycistronic genes, for instance, the *stv1* mutant in *Arabidopsis*, one of the *RPL24* genes, through perturbation of translation reinitiation of the auxin response factor (*ARF*) transcripts caused the defects in gynoecium patterning [52]. Thus, we assume that these two candidate genes may promote the growth rate of leaf thickness and perturb translation reinitiation of *ARF* genes during the leaf number-changing process in slipper orchids, respectively, and are worthy of further investigation.

## 5. Conclusions

In this study, a high-density genetic map for lady's slipper orchids was constructed with 8410 SNP markers using the RNA sequencing technique. The map spanned 1616.18 cM with an average distance of 0.19 cM between adjacent markers. Furthermore, this map was used to identify QTLs of four leaf-related traits, including leaf length, leaf width, leaf thickness, and leaf number. Finally, 12 QTLs distributed on 8 LGs were identified. From these QTLs regions, 25 candidate genes controlling four leaf-related traits were identified. More studies are needed to explore their potential roles and functions in leaf length, leaf width, leaf thickness, and leaf number formation in the future. In conclusion, the high-density genetic map, QTLs, and candidate genes studied here have provided useful tools for marker-assisted selection breeding of leaf-related traits in lady's slipper orchids, and further map-based cloning, comparative genomes analyses, and whole-genome assembly.

**Supplementary Materials:** The following supporting information can be downloaded at: https://www.mdpi.com/article/10.3390/horticulturae8090842/s1. Table S1: Leaf length, leaf width, leaf thickness, and leaf number statistics of *P. concolor*, *P. hirsutissimum*, and their F1 population collected in 2018. Table S2: The results of RNA detection. Table S3: Statistical for sample RNA sequencing data evaluation. Table S4: Annotation of unigenes to the publicly available databases. Table S5: SNP markers segregation information used for the final integrated map. Table S6: The SNP markers distributed on 13 linkage groups in the integrated map and their corresponding unigenes. Table S7: The SNP markers distributed on 13 linkage groups in the female map and their corresponding unigenes. Table S8: The SNP markers distributed on 13 linkage groups in the male map and their corresponding unigenes. Table S9: Phenotypic variance explained values correspond to four leaf-related traits QTLs associated with each SNP marker. Table S10: Information and annotation of 38 SNP markers located in the four leaf-related traits QTLs. Table S11: GO enrichment analysis of candidate genes located in four leaf-related traits QTLs regions. Figure S1: Comparison of second leaf morphologies from the top between *P. concolor* (up) and *P. hirsutissimum* (down). Figure S2: Distribution of SNP markers on the 13 linkage groups of female parent *P. concolor*. Figure S3: Distribution of SNP markers on the 13 linkage groups of male parent *P. hirsutissimum*. Figure S4: Distribution of marker integrity on each individual in the F1 mapping population. The *x*-axis indicates each of the F1 individuals, and the *y*-axis indicates the integrity of mapping markers in each individual in the F1 mapping population. File S1: FASTA formatted file of unigene sequences from the assembled transcriptome. File S2: The haplotype maps of each individual in 13 LGs by using 8410 SNP markers. File S3: The heat maps regarding the recombination relationship between 8410 mapped SNP markers on each LG.

**Author Contributions:** D.-M.L. designed the project, developed the slipper orchids' F1 mapping population, planted and collected the leaf traits, constructed the genetic map and QTLs map, analyzed the data, and wrote the manuscript. G.-F.Z. conceived the study. All authors have read and agreed to the published version of the manuscript.

**Funding:** This research was supported by the National Natural Science Foundation of China (No. 31501788), Guangdong Basic and Applied Basic Foundation Project (No. 2021A1515010893), and the High-level of Guangdong Agricultural Science and Technology Demonstration City Project (2022). The funding bodies have no role in the study design, data collection and analysis, decision to publish, or preparation of the manuscript.

**Data Availability Statement:** The sequenced F1 population datasets presented in this study can be found in online repositories. The names of the repository/repositories and accession number (s) can be found below: https://www.ncbi.nlm.nih.gov (accessed on 5 September 2022), PRJNA760286. Other data presented in the current study are available on request from the corresponding author.

**Acknowledgments:** The authors are very thankful for the anonymous reviewer's suggestions for improving the manuscript.

**Conflicts of Interest:** The authors declare no conflict of interest.

## Abbreviations

| | |
|---|---|
| bp | base pairs |
| BLAST | Basic Local Alignment Search Tool |
| NCBI | National Center for Biotechnology Information |
| DNA | Deoxyribonucleic Acid |
| RNA | Ribonucleic acid |
| SNPs | Single Nucleotide Polymorphisms |
| QTL | Quantitative Trait Loci |
| LG | Linkage Group |
| GO | Gene Ontology |
| KEGG | Kyoto Encyclopedia of Genes and Genomes |
| COG | Cluster of Orthologous Groups |

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
