# Peer review of "High-Density Genetic Linkage Map Construction and QTLs Identification Associated with Four Leaf-Related Traits in Lady’s Slipper Orchids (Paphiopedilum concolor × Paphiopedilum hirsutissimum)"

_horticulturae, doi:10.3390/horticulturae8090842_

Round 1
Reviewer 1 Report
The manuscript presented is of good quality and can be considered for publication.
Here are my suggestions and comments to the authors for improvement.
for how many years the 95 f1 were phenotyped ( if this is for one year the QTL detection may not be so reliable).
For the RNA please explain if biological replication was considered
If the authors conducted genome assembly and later used GATK for the NAP calling how accurate is this as no reference genome was mentioned to be used? Why not use the Universal Network Enabled Analysis Kit (UNEAK) this is more accurate. The species is diploid 2n = 26 so how come we have double and triple annotation SNP.
The genetic map construction of how the integrated SNP was constructed was not well described and more details should be provided. Also, please give the recombination fraction (RF) across the LD. Any reason why a high gap was obtained on the LG11 please clarify. R/QTL is not so good in detecting the phenotypic variance explained (PVE) by the associated QTL please consider using step-wise regression to see the PVE. In R/QTL and Using permutation, the package will detect the QTL above the threshold, why do the authors decide to set this as manual? this does not sound good. If no QTL was detected using the permutation you could try Bonferroni method.
Reviewer 2 Report
An interesting research paper on a specialised topic. Much consideration has been given to leaf morphology as a complex process. So is the process of flowering. Some qualitative information on flowering should be included such as shape, colour and size in the introduction.
Also as the medicinal properties of such orchids has been noted, the part/parts of the plant is used eg flower, leaf, roots, stems or whole plant.
In future the genetics of the leaves could be of interest to the pharmacological components of leaves.
Editing aspects - in my printed copy -
• for graphic information some basic information should be added on how to interpret the chromosome information on p 12/13
• Table 6 - position of leaf traits needs to better correspond
• Table 7 - the word Marker needs realigning
Reviewer 3 Report
In the manuscript “High-Density Genetic Linkage Map Construction and QTLs Identification Associated with Four Leaf-related Traits in Lady’s Slipper Orchids (Paphiopedilum concolor ×Paphiopedilum hirsutissimum)”, the authors generate an interspecific F1 population from two economically important Paphiopedilum species and identify candidate genes affecting various leaf traits. The methods employed here represent modern approaches to non-model quantitative genetics. The manuscript is overall well-written and does an excellent job providing numerous figures and tables describing the population and genetics.
Materials and Methods:
2.2: by "normal distributions" do you mean tests for normality? If so, which test exactly? Alternatively, you may mean that you fit the data with a normal density curve (as seen in figures), which does not necessarily indicate that the data are normally distributed.
2.4: Not clear to me if the transcriptome annotation followed/used Trinotate or just a custom procedure. If you used Trinotate, please indicate in the text.
2.4: You talk about clean reads (and later clean data), but I don't see where you describe the software used for cleaning reads.
2.4: In addition to software not described, what versions/parameters were used for your programs? Default parameters?
2.4: Depending on the version and parameters used when running Trinity, your resulting assembly may vary drastically. Notably, you have many more contigs than biologically expected. Newer versions of Trinity perform in silico normalization to reduce assembly sizes to something better representative of the true transcriptome. The use of unigenes here somewhat helps with that process, but good preprocessing would've been better.
2.5: Instead of " two-allelic coding rule in genetics", you could just say ", and we used biallelic variants". Similarly, in Table 4/section 3.3/etc. you say "Bio-allelic", when you mean biallelic.
2.7: You did permutation tests for LOD estimation, but that step typically provides an adjusted LOD threshold (not just 3, which is a standard default).
2.7: You cite [36] for the estimation of PVE, but it would be better to provide the formula with the citation. It appears to be "1-10^(-(2/n)*LOD)"
2.7: As for PVE in this population, the number of samples is low. You should probably provide a disclaimer for PVE estimates.
See King, E. G., & Long, A. D. (2017). The Beavis effect in next-generation mapping panels in Drosophila melanogaster. G3: Genes, Genomes, Genetics, 7(6), 1643-1652.
General Comments:
I didn’t see any heritability estimates for your traits.
It's not clear whether detected QTL are likely to be associated with maturity. Often, such phenotypic traits are mapped with some measure of maturity as a covariate. As Paphiopedilum have long juvenile periods, the effects of maturity loci are likely to have a strong effect on such traits and not just those traits but many agronomic/biomass traits. There should be some mention of maturity as either a negligible factor or something for future studies to consider.
I’m curious if you’ve considered using this population for a transcriptome-wide association study. It seems like you could attempt one given your current data set.
There is various highlighting throughout the manuscript that should be removed.
It would be nice to include the genotypic data used (preferably in R qtl format). Similarly, scripts used to analyze the data would further benefit the reproduction of these results.
Figure/Table Comments:
File S1 legend - "FASTA formatted file of unigene..." not "Faster form"
Figures 1 and 4 – subplots (A, B, C, and D) are different sizes and not lined up properly.
Figure 4 - No legend?
Figure S4 - something odd happened with this figure. It's entirely green.
Figure S5 - this figure isn't really worth including. If you want to show variation of haplotypes/allele frequencies/recombination break points, there are better ways to do that.
Figure S6 is quite large. You may want to consider reformatting to a smaller size.
Tables S4, S7, and S8 were giving me file format issues. Please, double check that the files are properly formatted. It would be best to consistently use “XSLX” format.
Round 2
Reviewer 1 Report
The authors have addressed my concerns even though I am still having little. doubt on the pairwise provided for each linkage. It seems markers were not properly ordered.